# Multi-Institutional Comparison of Ablative 5-Fraction Magnetic Resonance-Guided Online Adaptive Versus 15/25-Fraction Computed Tomography-Guided Moderately Hypofractionated Offline Adapted Radiation Therapy for Locally Advanced Pancreatic Cancer

**DOI:** 10.3390/cancers17152596

**Published:** 2025-08-07

**Authors:** Michael D. Chuong, Eileen M. O’Reilly, Robert A. Herrera, Melissa Zinovoy, Kathryn E. Mittauer, Muni Rubens, Adeel Kaiser, Paul B. Romesser, Nema Bassiri-Gharb, Abraham J. Wu, John J. Cuaron, Alonso N. Gutierrez, Carla Hajj, Antonio Ucar, Fernando DeZarraga, Santiago Aparo, Christopher H. Crane, Marsha Reyngold

**Affiliations:** 1Department of Radiation Oncology, Miami Cancer Institute, Miami, FL 33176, USA; robertoherr@baptisthealth.net (R.A.H.); kathrynm@baptisthealth.net (K.E.M.); adeelk@baptisthealth.net (A.K.); nema.bassirigharb@baptisthealth.net (N.B.-G.); alonsog@baptisthealth.net (A.N.G.); 2Department of Medical Oncology, Memorial Sloan Kettering Cancer Center, New York, NY 10065, USA; oreillye@mskcc.org; 3Department of Radiation Oncology, Memorial Sloan Kettering Cancer Center, New York, NY 10065, USA; remism@mskcc.org (M.Z.); romessep@mskcc.org (P.B.R.); wua@mskcc.org (A.J.W.); cuaronj@mskcc.org (J.J.C.); hajjc@mskcc.org (C.H.); cranec1@mskcc.org (C.H.C.); reyngolm@mskcc.org (M.R.); 4Office of Clinical Research, Miami Cancer Institute, Miami, FL 33176, USA; munir@baptisthealth.net; 5Department of Medical Oncology, Miami Cancer Institute, Miami, FL 33176, USA; antoniou@baptisthealth.net (A.U.); fernandoz@baptisthealth.net (F.D.); santiagoap@baptisthealth.net (S.A.)

**Keywords:** pancreatic cancer, ablative radiation therapy, SMART, magnetic resonance imaging (MRI), image-guided radiotherapy, MR-guided radiotherapy, CT-guided radiotherapy, adaptive radiation therapy

## Abstract

We conducted the first comparison of dose-escalated 5-fraction SMART and 15/25-fraction HART for locally advanced pancreatic cancer. Despite a higher proportion of tumors treated with SMART being located in the pancreatic head and presenting with anatomically unfavorable features, target volume coverage and hotspot intensities were greater in SMART plans, potentially facilitated by the use of online adaptive radiation therapy. Two-year local failure was significantly lower with SMART compared to HART, although two-year overall survival did not differ significantly. Both dose-escalated SMART and HART achieved favorable oncologic outcomes and should be evaluated in future prospective trials.

## 1. Introduction

Pancreatic ductal adenocarcinoma (PDAC) is projected to be the second leading cause of cancer-related death in the United States by 2030 [1,2]. Radiation therapy (RT) remains controversial for locally advanced pancreatic cancer (LAPC) [3]. Randomized controlled trials have not demonstrated an overall survival (OS) benefit with the addition of RT versus chemotherapy alone, thus leading to debate about its optimal utilization [4,5]. However, prior trials have routinely limited the prescribed biologically effective dose (BED)_10_ to ~60–70 Gy_10_, owing to the proximity of pancreatic tumors to gastrointestinal (GI) organs at risk (OARs) [6]. 

Recent studies suggest that ablative radiation therapy (A-RT) for LAPC can be delivered safely using advanced techniques that incorporate motion management and novel image guidance strategies, which might improve both LC and OS [7]. Two of the most commonly reported A-RT techniques are 5-fraction stereotactic magnetic resonance-guided online adaptive radiation therapy (SMART) [8,9,10,11,12,13,14] and 15/25-fraction computed tomography (CT)-guided moderately hypofractionated ablative radiation therapy (HART) [15,16]. Two-year OS after induction chemotherapy and A-RT was 40.5% [9] in the multi-center phase 2 SMART trial and 38% after HART in two single-institution retrospective analyses, while the historical two-year OS from the time of diagnosis with chemotherapy +/− non-ablative RT is ~25% [4,5,16,17]. 

While oncologic outcomes after both SMART and HART are promising, these treatment strategies have not previously been compared, leading to uncertainty about whether there are meaningful differences. 

## 2. Materials and Methods

### 2.1. Study Design and Participants

This multi-center retrospective cohort study included consecutive patients from two high-volume PDAC institutions in the United States. Institutional review board (IRB) approval was obtained from both institutions and a waiver of informed consent was granted as this was a minimal-risk study (IRB approval: 1880120 and 16-370A).

All patients had histologically confirmed LAPC and were treated either with 5-fraction SMART or 15/25-fraction HART without surgery as per institutional standard of care. SMART was exclusively used for patients treated at the Miami Cancer Institute (MCI; Miami, FL, USA). while HART was used for all patients from the Memorial Sloan Kettering Cancer Center (MSKCC; New York, NY, USA), where an MR-Linac was not yet available. Patients were included regardless of induction chemotherapy status, CA19-9 level, and tumor size. Patients with distant metastases prior to RT were excluded, as were patients who had prior abdominal RT and definitive surgery after A-RT.

Initial staging was typically performed with pancreatic protocol contrast-enhanced CT scans of the chest, abdomen, and pelvis at the MSKCC, which offered HART, while a CT scan of the chest and MRI scan of the abdomen was routine at the MCI, which treated with SMART. Staging positron emission tomography (PET)/CT scans were rarely used at both institutions.

### 2.2. Five-Fraction SMART

The SMART planning and delivery approach has been previously published [18]. Briefly, SMART was delivered on a 0.35 Tesla (T) MR-Linac (ViewRay, Oakwood Village, OH, USA) using a step-and-shoot approach typically with ~17–20 beams at various angles and ~40–60 total segments [19]. MR simulation was performed using the MR-Linac without contrast and without fiducial markers. Treatment was usually delivered with continuous intrafraction multi-planar cine-MRI and automatic beam gating, typically with mid-inspiration breath hold (BH). Online adaptive replanning was performed when there was a predicted GI OAR constraint violation based on the original plan’s fluence imposed on the patient’s anatomy on each treatment day.

Nearly all SMART patients (*n* = 87; 95.6%) were prescribed 50 Gy in 5 fractions (BED_10_ = 100 Gy_10_); 4 (4.4%) were prescribed 45 Gy in 5 fractions (BED_10_ = 85.5 Gy_10_). The GTV included the primary tumor and clinically involved lymph nodes. A microscopic clinical target volume (CTV_microscopic_) was routine and included an approximately 5–10 mm margin around the gross tumor volume (GTV), celiac artery (CA), and superior mesenteric artery (SMA); the porta hepatis and/or para-aortic regions were included based on physician preference. The CTV was prescribed 33 Gy in 5 fractions using a simultaneous integrated boost. The PTV50 and PTV33 were created from an isotropic 3 mm expansion of the GTV and CTV_microscopic_, respectively. A 3 mm GI planning OAR volume (PRV) was standard. Select dose constraints include the following: stomach/duodenum/small bowel V35Gy < 0.5 cc, V40Gy < 0.03 cc; large bowel V38Gy < 0.5 cc, V43Gy < 0.03 cc [10]. An isotoxicity planning approach was used, by which plans were normalized to a GI OAR constraint to optimize target coverage. There was no formal maximum hotspot constraint, with hotspots intended to be ~130–140% of the prescribed dose.

SMART patients did not receive concurrent chemotherapy or prophylactic proton pump inhibitors (PPIs).

### 2.3. Fifteen/Twenty-Five-Fraction HART

HART was delivered on a conventional linac (Varian Medical Systems, Palo Alto, CA, USA) using volumetric modulated arc therapy (VMAT), as previously described [7]. Briefly, prior to simulation, patients without metal stents or surgical clips underwent fiducial placement. CT simulation was performed using intravenous contrast unless contraindicated with RPM-mediated motion management. Motion management consisted of deep inspiration breath hold when tolerable or end-expiratory gating. Treatment was delivered using kilovoltage (kV) X-ray stereoscopic planar images and cone-beam CT (CBCT) for image guidance with intermittent kV intrafraction motion monitoring.

If there was no abutment of the GTV with luminal GI OARs, then the prescribed dose to gross disease and an elective volume was 67.5 Gy (BED_10_ = 97.88 Gy_10_) and 37.5 Gy (BED_10_ = 46.88 Gy_10_) in 15 fractions, respectively. Otherwise, the prescribed dose to gross disease and an elective volume was 75 Gy (BED_10_ = 97.5Gy_10_) and 45 Gy in 25 fractions (BED_10_ = 53.1 Gy_10_), respectively. 

The GTV contour included the primary tumor and clinically involved lymph nodes. A mandatory CTV_microscopic_ was created from a 10 mm expansion around the GTV, CA, and SMA, as well as any vessels involved by the tumor, +/− the portal vein and splenic hilum for select tumors based on location. The PTV encompassing gross disease was created using a 5 mm isotropic expansion and subsequently excluding GI OARs with an extra 5–7 mm margin. The PTV encompassing the elective volume was created by adding a 5 mm isotropic margin to the CTV_microscopic_. A 3–5 mm GI OAR PRV was standard. Fractionation-dependent GI OAR constraints have been previously published [7]. For the 15-fraction regimen, stomach/duodenum/small bowel constraints included D_max_ < 45 Gy and V_37.5Gy_ < 40 cc, as well as D_5cc_ of the corresponding PRV < 45 Gy. For the 25-fraction regimen, stomach/duodenum/small bowel constraints included D_max_ < 60 Gy, V_45Gy_ < 40 cc and D_5cc_ of the corresponding PRV < 60 Gy. The maximum PTV dose was typically constrained to 115–120% of prescription, unless a deliberate 130% hotspot was selectively placed inside large tumors (N-17, 14.2%), as previously described [7]. 

Weekly CBCT review was performed to estimate the percentage of time that luminal GI OARs overlapped with the 45 Gy and 60 Gy isodose lines (IDLs) for 15- and 25-fraction regimens, respectively. Selective replanning was performed halfway through the treatment course if OARs were overlapped by these critical IDLs more than 33% of the time, and this was performed for 11 patients (9.2%).

Nearly all patients treated with HART received concurrent capecitabine at 825 mg/m^2^ twice daily on the days of HART (*n* = 117, 97.5%). Prophylactic PPIs or H2 receptor antagonists were selectively used.

### 2.4. Additional Therapy After SMART/HART

The use of maintenance chemotherapy was at the discretion of the treating oncologist. No patient had definitive surgery for PDAC after SMART or HART.

### 2.5. Response Assessment After SMART/HART

The follow-up protocol at both institutions was similar and included physical examination, diagnostic imaging studies, and labs including CA19-9 every 3 months after A-RT. Diagnostic imaging studies typically consisted of CT and/or MRI, while PET/CT scans were occasionally ordered to clarify equivocal CT/MRI scan and CA19-9 changes.

### 2.6. Toxicity

Acute and late adverse events were defined as per Common Terminology Criteria for Adverse Events (CTCAE) version 5.0 and occurred within or after 3 months of RT, respectively.

### 2.7. Statistical Analysis

A comparison of demographic and clinical characteristics between the SMART and HART cohorts was conducted. Categorical and continuous variables were compared by a chi-square test and *t*-test, respectively.

Local failure (LF) and distant failure (DF) were estimated by the Fine and Gray method, with death as a competing risk and compared between the SMART and HART cohorts by the Gray’s test. OS and progression-free survival (PFS) were estimated by the Kaplan–Meier method and compared between the SMART and HART cohorts by the log-rank test. These analyses were performed from the start of SMART/HART. To account for potential imbalances between the SMART and HART cohorts, we attempted to perform propensity score matching.

LF was defined by the Response Evaluation Criteria in Solid Tumors (RECIST) version 1.1 using the diagnostic scan reports. Gray’s test and competing risk regression was used to identify factors associated with LF and DF. Cox proportional regression was used to identify factors associated with OS. Variables with a *p* value < 0.1 in the univariate analysis (UVA) were included in the final multivariate analysis (MVA). Variance inflation factor greater than 10 were set as the cut-off for multicollinearity and excluded from the models. All tests were two-sided, and statistical significance was set at *p* < 0.05.

## 3. Results

A total of 211 patients (SMART, *n* = 91; HART, *n* = 120) were evaluated. Patients were treated with SMART and HART in 2018–2023 and 2016–2019, respectively. Baseline patient and tumor characteristics are described in Table 1. Tumors treated with SMART were more often located in the head of the pancreas (80.2% vs. 62.5%; *p* = 0.005) and had a higher volume of GI OARs located within isotropic 3 mm (median 1.4 vs. 1.0 cc; *p* = 0.036) or 5 mm (3.9 vs. 2.8 cc; *p* = 0.028) expansions around the GTV, respectively. Despite this, SMART plans were hotter and had higher target coverage based on comparison of the original plans (Table 2).

Median follow-up for SMART/HART was 27.0 vs. 40.0 months (*p* < 0.0002), respectively. Two-year cumulative incidence of LF and DF from SMART vs. HART was 6.5% vs. 32.9% (*p* < 0.001) and 68.3% vs. 71.5% (*p* = 0.576), respectively (Figure 1A,B). After matching for “GTV + 5mm overlap with GI OARs” and “N stage”, there was an insufficient number of patients to evaluate whether a significant LF difference existed due to sparse data bias. Median PFS and OS from SMART/HART were 7.0 vs. 5.0 months (*p* = 0.792), and 14.0 vs. 17.0 months (*p* = 0.217), respectively (Table 3). Two-year PFS and OS for SMART vs. HART were 15.9% vs. 14.1% (*p* = 0.991), and 31.0% vs. 35.3% (*p* = 0.056), respectively (Figure 1C,D). After matching for “age “and “N stage”, we found no statistically significant difference in OS.

Dosimetric characteristics of the original (not adapted) SMART and HART plans associated with LF (*n* = 36) vs. no LF (*n* = 171) are shown in Table 4. There was no difference in GTV or proximity to GI OARs. Higher GTV coverage and plans with higher hotspots were more common among patients with no LF.

Significant factors associated with LF on MVA included use of HART vs. SMART (reference) (hazard ratio [HR] 5.389, 95% confidence interval [CI] 1.298–21.975; *p* = 0.021) and non-mFOLFIRINOX vs. mFOLFIRINOX induction (reference) (HR 2.067, 95% CI 1.038–4.052; *p* = 0.046 (Table 5). Several dosimetric parameters from the original plans, as shown in Table 5, were significantly associated on UVA but not on MVA. Factors associated with OS on MVA included post-induction CA19-9 decrease > 40% vs. ≤40% (reference) (HR 0.725, 95% CI 0.515–0.996; *p* = 0.046) and GTV V120% as a continuous variable (HR 1.022, 95% CI 1.006–1.037; *p* = 0.015).

The incidence of acute grade 3 toxicity was similar in the SMART and HART cohorts (3.3% vs. 5.8%; *p* = 0.390) and included bowel obstruction/stenosis (*n* = 6), GI bleeding (*n* = 3), and abdominal pain (*n* = 1). There was no acute grade 4–5 toxicity. Late grade > 3 toxicity was less common among patients treated with SMART (2.2% vs. 9.2%; *p* = 0.037). Late grade 3 toxicities included GI bleeding (*n* = 10), gastric obstruction (*n* = 1), and duodenitis (*n* = 1). One late grade 4 adverse event (duodenal perforation) occurred in a patient treated with HART, while there was no late grade 4–5 toxicity in the SMART cohort. No grade > 3 acute or late toxicity occurred in HART patients who underwent selective offline adaptive replanning.

## 4. Discussion

There has been growing enthusiasm about the potential for radiation dose escalation to improve outcomes for LAPC patients although technological limitations have restricted the routine use of ablative RT [20,21,22,23,24]. 

Two novel treatment strategies for delivering ablative radiation dose for LAPC have resulted in encouraging clinical outcomes. In 2016, Krishnan and colleagues at the MD Anderson Cancer Center first published retrospective HART outcomes most commonly in 15 or 28 fractions with CT-on-rails or CBCT guidance and concurrent capecitabine [16]. Patients prescribed a BED_10_ > 70 Gy_10_ had improved OS (2-year: 36% vs. 19%, 3-year: 31% vs. 9%). The Memorial Sloan Kettering Cancer Center’s retrospective outcomes using a similar approach in 15 or 25 fractions (both having a prescribed BED_10_ ~98 Gy_10_) with CBCT, selective offline adaptive replanning, and concurrent capecitabine included a 2-year LF and OS after HART of 32.8% and 38%, respectively, and acceptable incidence of grade ≥ 3 toxicity [17]. On the other hand, SMART features MRI guidance, continuous intrafraction imaging, automatic beam gating, and online adaptive replanning [25]. Early studies suggested that these advanced capabilities could allow the ablative dose to be safely delivered in five fractions [26]. Recently published results of the multi-center phase 2 SMART trial (*n* = 136) that prescribed 50 Gy in five fractions (BED_10_ = 100 Gy_10_) provide prospective confirmation that SMART is safe; the primary endpoint of acute grade ≥ 3 GI toxicity definitely attributed to SMART was met, with the reported incidence being 0% [8]. In an updated analysis, the study investigators reported an encouraging 2-year OS of 40.5% for SMART, and the incidence of late grade ≥ 3 toxicity probably or definitely related to SMART was < 5% [9].

An unanswered question is whether clinical outcomes for LAPC differ between 5-fraction SMART and 15/25-fraction HART. The current analysis is, to the best of our knowledge, the first comparison of dosimetric and oncologic outcomes between these treatment approaches. Whereas prior non-ablative studies have consistently reported 2-year LF ~50% [27,28], herein, we report 2-year LF rates that are among the lowest in the published literature for PC (SMART: 6.5% vs. HART: 32.9%; Gray’s *p* < 0.001). We believe that our analysis is clinically meaningful as it addresses an outstanding question about optimal technology triage when dose escalating RT for LAPC, especially considering that few centers currently have an MR-Linac, while HART can be performed at most centers that offer advanced motion management on a conventional linac. 

Our evaluation of SMART and HART treatment plans showed that patients who experienced LF had inferior GTV coverage than those who did not have LF, especially with respect to D_90%_ and D_80%_. Moreover, treatment plans associated with LF had lower hotspots, but not a lower D_min_, suggesting treatment techniques that can deliver higher dose to more of the GTV may decrease LF. Interestingly, treatment technique (SMART vs. HART) was a significant factor on MVA for LF, while dosimetric target coverage variables were not. It is likely that dosimetric parameters are not independent of technique, as a different set of planning procedures was used. In addition, we evaluated dosimetric outcomes of the original plans, and it is uncertain whether our results would have been different if the adapted plans had been evaluated, especially as the original SMART plans were rarely delivered. As such, we are unable to separate the effect of dosimetric differences from the effect of other aspects of treatment planning and delivery between SMART and HART, including fractionation scheme, quality of on-board imaging, and frequency of online adaptive replanning.

There are several novel components of SMART delivered on a 0.35T MR-Linac, including MRI guidance and online adaptation, although the current study was not designed to comprehensively evaluate the relative benefits of each compared to HART. Nonetheless, it is plausible that daily online adaptive replanning might have facilitated a lower incidence of LF by optimizing target coverage at least in part by supporting a more permissive set of constraints. While online adaptive replanning is available on some CT-guided linacs [29], it is unclear whether using such technology achieves similar outcomes as we have reported with SMART. In addition, while the total prescribed BED_10_ was similar for SMART and HART, it is possible that our LF outcomes may have been influenced by radiobiologic differences related to fraction size; enhanced indirect cell killing mechanisms, such as endothelial cell apoptosis mediated by the sphingomyelin pathway and tumor vascular damage, may be triggered by fraction sizes of at least 10 Gy, which was routine for SMART [30,31,32].

While elective volume coverage is not common for treating LAPC, especially with stereotactic body radiation therapy, SMART and HART patients were routinely treated with an anatomically derived CTV [33,34,35]. Elective volume coverage might have contributed to achieving favorable LF rates, at least in part since the true extent of PDAC is poorly represented on imaging studies, resulting in the potential for the GTV to be undercontoured [36]. In addition, PDAC is well known to have occult microscopic extension along perineural tracks and within regional lymph nodes, resulting in a high rate of marginal and out-of-field recurrence after targeting only the gross tumor [37,38]. A contouring atlas was recently published for the upcoming phase 3 NRG GI011 randomized trial that will require elective coverage in the dose-escalated RT arm [39].

We did not observe a statistically significant difference in median OS (14.0 vs. 17.0 months; *p* = 0.217) or 2-year OS (31.0% vs. 35.3%; *p* = 0.056) after SMART compared to HART. The lower LF achieved with SMART could have hypothetically impacted OS, either by decreasing deaths from uncontrolled local tumor growth or reducing the number of cells with metastatic potential [40]. However, it is possible that we did not detect an OS difference as various patient, tumor, and treatment factors also impact OS. Nonetheless, we believe that the promising long-term OS achieved after both SMART and HART supports further prospective evaluation of radiation dose escalation for LAPC. 

We recognize several limitations of this study, especially its retrospective design, which introduces the potential for underreporting of toxicity and selection bias. Institutional differences in planning techniques are likely to have influenced dosimetric outcomes irrespective of treatment technique, and dosimetric evaluation was not performed on the adapted plans. Furthermore, institutional differences in the staging, systemic therapy delivery, and patient support pathways may have contributed to the differences in outcomes.

## 5. Conclusions

In conclusion, this analysis adds to the published literature by demonstrating that SMART and HART can safely deliver an ablative dose with routine elective coverage and achieve favorable LF and OS rates for select patients with LAPC following induction chemotherapy. The data reported herein support either SMART or HART for treatment of LAPC. The preferred technique may, in large part, depend on the available technology and experience at each institution. Although differences in dosimetric and clinical outcomes were observed, these findings should be considered hypothesis-generating and support future prospective evaluation. We encourage enrollment to clinical trials that aim to clarify the role of A-RT for LAPC.

## Figures and Tables

**Figure 1 cancers-17-02596-f001:**
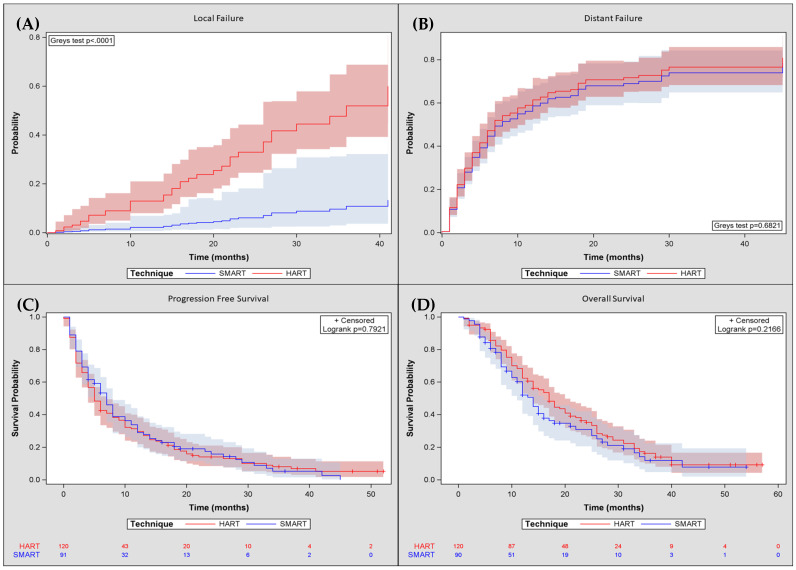
Cumulative incidence of local (**A**) and distant failure (**B**) and Kaplan–Meier plots for progression-free survival (**C**) and overall survival (**D**) from the start of stereotactic magnetic resonance-guided online adaptive radiation therapy (SMART) or computed tomography-guided moderately hypofractionated ablative radiation therapy (HART).

**Table 1 cancers-17-02596-t001:** Baseline patient, tumor, and treatment characteristics.

	SMART (*n* = 91)	HART (*n* = 120)	Total (*n* = 211)	*p* Value
**Age (year), median**	72.0 (47–94)	68.0 (42–91)	70.0 (42–94)	0.004
**Gender** Male Female	49 (53.8%) 42 (46.2%)	59 (49.2%) 61 (50.8%)	108 (51.2%) 103 (48.8%)	0.501
**ECOG PS** 0–1 2	85 (93.4%) 6 (6.6%)	108 (90.0%) 12 (10.0%)	193 (91.5%) 18 (8.5%)	0.380
**Tumor location** Head Body/tail	73 (80.2%) 18 (19.8%)	75 (62.5%) 45 (37.5%)	148 (70.1%) 63 (29.9%)	0.005
**T stage** 1–2 3–4	23 (25.3%) 68 (74.7%)	20 (16.7%) 100 (83.3%)	43 (20.4%) 168 (79.6%)	0.124
**N stage** Positive Negative Unknown	23 (25.3%) 68 (74.7%) 0 (0.0%)	51 (42.5%) 54 (45.0%) 15 (12.5%)	74 (35.1%) 122 (57.8%) 15 (7.1%)	< 0.001
**Tumor size (cm), median**	3.7 (1.5–6.9)	3.8 (1.4–7.4)	3.8 (1.4–7.4)	0.583
**Overlap w/GI OARs (cc), median** GTV + 3 mm GTV + 5 mm	1.4 (0–19.6) 3.9 (0–32.4)	1.0 (0–12.5) 2.8 (0–24.0)	1.2 (0–19.6) 3.2 (0–32.4)	0.036 0.028
**CA19-9 baseline (U/mL), median**	156.4 (1.0–19,000)	166.0 (0–2766)	165.0 (0–19,000)	0.780
**CA19-9 pre-RT (U/mL), median**	55.0 (1.2–11,534)	79.0 (0–3507)	70.0 (0–11,534)	0.909
**Induction chemotherapy duration (month), median**	3.9 (0–9.1)	3.7 (0–13)	3.8 (0–13)	0.531
**Induction chemotherapy regimen** FOLFIRINOX Gemcitabine/nab-paclitaxel Gemcitabine/cisplatin Other 5-fluorouracil-based Other gemcitabine-based Other None	49 (53.8%) 26 (28.6%) 6 (6.6%) 0 (0.0%) 1 (1.1%) 0 (0.0%) 9 (9.9%)	66 (55.0%) 37 (30.8%) 4 (3.3%) 4 (3.3%) 5 (4.2%) 1 (0.8%) 3 (2.5%)	115 (54.5%) 63 (29.9%) 10 (4.7%) 4 (1.9%) 6 (2.8%) 1 (0.5%) 12 (5.7%)	0.068
**Maintenance chemotherapy** Yes No	6 (6.6%) 85 (93.4%)	19 (15.8%) 101 (84.2%)	25 (11.8%) 186 (88.2%)	0.039

ECOG = Eastern Cooperative Oncology Group; PS = performance status; GI = gastrointestinal; OARs = organs at risk; SMART = stereotactic magnetic resonance-guided adaptive radiation therapy; HART = hypofractionated ablative radiation therapy.

**Table 2 cancers-17-02596-t002:** Dosimetric comparison between 5-fraction SMART and 15/25-fraction HART.

	SMART (*n* = 91)	HART (*n* = 120)	Total (*n* = 211)	*p* Value
**Prescribed BED_10_ (Gy)** 85.5 97.5 97.9 100	4 (4.4%) 0 (0.0%) 0 (0.0%) 87 (95.6%)	0 (0.0%) 97 (80.8%) 23 (19.2%) 0 (0.0%)	4 (1.9%) 97 (46.0%) 23 (10.9%) 87 (41.2%)	<0.001
**GTV (cc), median**	34.4 (5.6–155.6)	31.2 (2.2–206.7)	32.0 (2.2–206.7)	0.466
**GTV D_95%_ scaled (%), median**	91.5 (64.9–119.9)	76.6 (42.9–108.3)	82.7 (42.9–119.9)	<0.001
**GTV D_90%_ scaled (%), median**	101.1 (74.2–122.2)	84.5 (44.6–111.7)	91.2 (44.6–122.2)	<0.001
**GTV D_80%_ scaled (%), median**	106.7 (83.3–128.3)	95.2 (47.8–121.3)	101.0 (47.8–128.3)	<0.001
**GTV V_130%_ (cc), median**	2.8 (0–34.9)	0 (0–27.4)	0 (0–34.9)	<0.001
**GTV V_120%_ (cc), median**	14.5 (0–79.5)	0 (0–45.8)	0 (0–79.5)	<0.001
**GTV V_100%_ (cc), median**	29.4 (5.6–124.8)	20.5 (0–153.6)	25.4 (0–153.6)	<0.001
**GTV min dose scaled (%), median**	66.7	60.5	62.8	<0.001

BED = biologically effective dose; GTV = gross tumor volume; SMART = stereotactic magnetic resonance-guided adaptive radiation therapy; HART = hypofractionated ablative radiation therapy.

**Table 3 cancers-17-02596-t003:** Local failure, distant failure, progression-free survival, and overall survival from SMART/HART.

	Median (Months)					1-Year (95% CI)			2-Year (95% CI)		
	SMART		HART	*p* Value		SMART	HART	*p* Value	SMART	HART	*p* Value
**LF**	Not reached		36.8 (29.0–.)	**---**		3.1% (1.9–6.3%)	11.8% (9.5–15.4%	0.001	6.5% (2.5–14.6%)	32.9% (26.3–41.1%)	<0.001
**DF**	6.0 (5.0–10.0)	7.0 (6.0–14.0)	0.423	57.1% (51.2–62.2%)	59.1% (52.9–61.1%)	0.837	68.3% (61.2–74.1%)	71.5% (64.7–78.6%)			0.576
**PFS**	7.0 (5.0–8.0)	5.0 (4.0–7.0)	0.792	29.1% (19.4–38.7%)	29.8% (21.5–38.0%)	0.525	15.9% (7.8–24.0%)	14.1% (7.8–20.5%)			0.991
**OS**	14.0 (11.0–16.0)	17.0 (14.0–20.0)	0.217	52.0% (41.0–63.0%)	62.3% (53.5–71.1%)	0.272	31.0% (20.2–41.9%)	35.3% (26.5–44.2%)			0.056

LF = local failure; DF = distant failure; PFS = progression-free survival; OS = overall survival; SMART = stereotactic magnetic resonance-guided adaptive radiation therapy; HART = hypofractionated ablative radiation therapy; CI = confidence interval.

**Table 4 cancers-17-02596-t004:** Dosimetric differences between patients who experienced local failure versus no local failure.

	No Local Failure (*n* = 171)	Local Failure (*n* = 36)	*p* Value
**GTV (cc), median**	32	27.5	0.177
**GI OAR overlap (cc), median** GTV + 3 mm isotropic expansion GTV + 5 mm isotropic expansion	1.2 3.4	1 2.9	0.274 0.280
**GTV D_95%_ scaled (%), median**	83.9	77.7	0.103
**GTV D_90%_ scaled (%), median**	91.7	84.3	0.042
**GTV D_80%_ scaled (%), median**	101.7	84.8	0.003
**GTV V_130%_ (cc), median**	0.1	0	<0.001
**GTV V_120%_ (cc), median**	4.9	0	<0.001
**GTV V_100%_ (cc), median**	26.5	18.1	0.036
**GTV min dose scaled (%), median**	42	46.5	<0.001

GTV = gross tumor volume; GI = gastrointestinal; OAR = organ at risk.

**Table 5 cancers-17-02596-t005:** Univariate and multivariate analyses for local failure and overall survival after ablative radiation therapy.

	LF				OS			
Variables	Univariate		Multivariate		Univariate		Multivariate	
	HR (95% CI)	*p* Value	HR (95% CI)	*p* Value	HR (95% CI)	*p* Value	HR (95% CI)	*p* Value
**Age (year)**								
>Median	Reference				Reference			
≤Median	1.012 (0.525–1.902)	0.958			1.141 (0.836–1.556)	0.406		
**Gender**								
Male	Reference				Reference			
Female	1.085 (0.573–2.049)	0.826			1.068 (0.783–1.457)	0.678		
**ECOG performance status**								
0–1	Reference				Reference			
2	1.152 (0.537–1.983)	0.624			1.238 (0.726–2.109)	0.433		
**Tumor location**								
Body/tail	Reference				Reference			
Head	0.982 (0.512–1.874)	0.915			1.228 (0.874–1.725)	0.238		
**T stage**								
1–2	Reference				Reference			
3–4	1.150 (0.518–2.532)	0.751			0.977 (0.672–1.422)	0.905		
**N stage**								
N−	Reference				Reference			
N+	1.048 (0.535–2.049)	0.905			0.951 (0.683–1.324)	0.766		
N/A	1.345 (0.395–4.590)	0.645			1.020 (0.558–1.863)	0.949		
**Tumor size**								
>Median	Reference				Reference			
≤Median	0.762 (0.401–1.435)	0.361			0.904 (0.661–1.236)	0.527		
**GTV volume**								
>Median	Reference				Reference			
≤Median	1.099 (0.578–2.102)	0.790			1.279 (0.938–1.744)	0.119		
**Radiation technique**								
5-fraction SMART	Reference				Reference			
15/25-fraction HART	6.510 (1.975–21.350)	0.001	5.389 (1.298–21.975)	0.021	0.823 (0.598–1.132)	0.23		
**CA19-9 prior to RT**	1.000 (1.000–1.000)	0.521			1.000 (1.000–1.000)	0.204		
**CA19-9 > 40% decrease prior to RT**								
No	Reference				Reference			
Yes	1.105 (0.550–2.218)	0.798			0.675 (0.481–0.947)	0.023	0.703 (0.500–0.987)	0.042
**Induction chemo regimen**								
FOLFIRINOX	Reference				Reference			
Others	2.104 (1.067–4.098)	0.032	2.067 (1.038–4.052)	0.047	1.136 (0.833–1.548)	0.42		
**Induction chemo duration**								
>Median	Reference				Reference			
≤Median	1.462 (0.735–2.901)	0.289			0.787 (0.575–1.078)	0.136		
**GTV D_80%_ scaled (%), median**	0.971 (0.948–0.996)	0.031	0.971 (0.902–1.038)	0.376	1.003 (0.991–1.014)	0.625		
**GTV V_120%_ (cc), median**	0.889 (0.815–0.978)	0.019	0.992 (0.936–1.054)	0.553	1.021 (1.009–1.034)	0.001	1.021 (1.007–1.034)	0.002
**GTV minimum dose**	0.997 (0.976–1.020)	0.710			0.995 (0.983–1.006)	0.366		

LF = local failure; OS = overall survival; GTV = gross tumor volume; HR = hazard ratio; CI = confidence interval; ECOG = Eastern Cooperative Oncology Group; SMART = stereotactic magnetic resonance-guided adaptive radiation therapy; HART = hypofractionated ablative radiation therapy.

## Data Availability

Research data are stored in an institutional repository and will be made available upon request to the corresponding authors.

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
