# Peer review of "Multi-Institutional Comparison of Ablative 5-Fraction Magnetic Resonance-Guided Online Adaptive Versus 15/25-Fraction Computed Tomography-Guided Moderately Hypofractionated Offline Adapted Radiation Therapy for Locally Advanced Pancreatic Cancer"

_cancers, 2025, doi:10.3390/cancers17152596_

Round 1
Reviewer 1 Report
Comments and Suggestions for Authors
This multi-institutional retrospective study compares two dose-escalated ablative radiation therapy strategies for treating locally advanced pancreatic cancer (LAPC): 5-fraction magnetic resonance-guided online adaptive radiation therapy (SMART) and 15/25-fraction computed tomography-guided offline adaptive moderately hypofractionated radiation therapy (HART). Among 211 patients, SMART demonstrated superior gross tumor volume (GTV) coverage and higher hotspot intensities. Despite a higher proportion of anatomically unfavorable tumors in the SMART cohort, two-year local failure (LF) was significantly lower in SMART compared to HART, although no significant difference in overall survival (OS) was observed. The findings suggest that SMART, through its daily adaptive planning and tighter dose distributions, may offer better local control, while both strategies remain effective and safe. The study supports the need for future prospective randomized trials to define the optimal approach. The minor revision have to be done.
- The study bases its dosimetric analysis only on the original, non-adapted treatment plans, while most SMART patients received daily adapted plans. Including data from adaptive plans would offer stronger support for the claimed benefits of SMART and more accurately reflect clinical practice.
- The retrospective design introduces potential biases, including institutional differences in planning and delivery. This limitation should be more prominently emphasized in the discussion, with a clearer call for prospective, randomized trials to validate these findings.
- The study includes patients with tumors located in various anatomical sites (head, body, tail) and differing levels of proximity to gastrointestinal organs at risk. Subgroup analyses exploring how these factors influence LF and OS would enhance the granularity and applicability of the results.
- Although the BED10 values are similar between SMART and HART, the biological effects of larger fraction sizes (e.g., endothelial apoptosis) could significantly influence treatment outcomes. A more detailed discussion of these mechanisms would strengthen the biological rationale.
- The Kaplan-Meier analyses for OS and PFS are presented with minimal interpretation. Given the borderline p-value for OS (P=0.056), a more thorough statistical assessment discussing clinical significance and sample size implications would be beneficial.
Author Response
This multi-institutional retrospective study compares two dose-escalated ablative radiation therapy strategies for treating locally advanced pancreatic cancer (LAPC): 5-fraction magnetic resonance-guided online adaptive radiation therapy (SMART) and 15/25-fraction computed tomography-guided offline adaptive moderately hypofractionated radiation therapy (HART). Among 211 patients, SMART demonstrated superior gross tumor volume (GTV) coverage and higher hotspot intensities. Despite a higher proportion of anatomically unfavorable tumors in the SMART cohort, two-year local failure (LF) was significantly lower in SMART compared to HART, although no significant difference in overall survival (OS) was observed. The findings suggest that SMART, through its daily adaptive planning and tighter dose distributions, may offer better local control, while both strategies remain effective and safe. The study supports the need for future prospective randomized trials to define the optimal approach. The minor revision have to be done.
Comments 1: The study bases its dosimetric analysis only on the original, non-adapted treatment plans, while most SMART patients received daily adapted plans. Including data from adaptive plans would offer stronger support for the claimed benefits of SMART and more accurately reflect clinical practice.
Response 1: Thank you for this comment, and we agree that evaluation of the cumulative radiation dose across the adapted plans would be especially useful. However, adaptive treatment planning systems do not report this and evaluating cumulative dose using 3rd party software is not feasible. We recognized this as a study limitation in our original submission: “dosimetric evaluation was not performed on the adapted plans.”
Comments 2: The retrospective design introduces potential biases, including institutional differences in planning and delivery. This limitation should be more prominently emphasized in the discussion, with a clearer call for prospective, randomized trials to validate these findings.
Response 2: We agree with these points and previously stated in the Discussion that potential bias and institutional differences could have influenced the study outcomes. We have emphasized these limitations. In the Conclusions, we previously stated that our data support future prospective evaluation. The phase 3 NRG GI011 trial includes both SMART and HART in the experimental arm although a prospective randomized trial comparing SMART vs. HART is unlikely to ever be conducted due to a high probability of poor accrual.
Original: “We recognize several limitations of this study including its retrospective design that introduces the potential for underreporting of toxicity and selection bias. Institutional differences in planning techniques could have influenced dosimetric outcomes irrespective of treatment technique and dosimetric evaluation was not performed on the adapted plans.”
Revised: “We recognize several limitations of this study especially its retrospective design that introduces the potential for underreporting of toxicity and selection bias. Institutional differences in planning techniques also likely influenced dosimetric outcomes irrespective of treatment technique and dosimetric evaluation was not performed on the adapted plans.”
Comments 3: The study includes patients with tumors located in various anatomical sites (head, body, tail) and differing levels of proximity to gastrointestinal organs at risk. Subgroup analyses exploring how these factors influence LF and OS would enhance the granularity and applicability of the results.
Response 3: While anatomic site can impact dosimetric outcomes and local failure risk, the imbalance in head vs. body/tail tumors between SMART and HART cohorts would likely prevent us from drawing meaningful conclusions if we attempted to perform additional subgroup analyses.
Comments 4: Although the BED10 values are similar between SMART and HART, the biological effects of larger fraction sizes (e.g., endothelial apoptosis) could significantly influence treatment outcomes. A more detailed discussion of these mechanisms would strengthen the biological rationale.
Response 4: We have expanded discussion about the radiobiological effects of fraction size as suggested.
Original: “In addition, while the total prescribed BED10 was similar for SMART and HART it is possible that our LF outcomes may have been influenced by radiobiologic differences related to fraction size; enhanced indirect cell killing mechanisms such as endothelial cell apoptosis may be triggered by fraction sizes of at least 10 Gy, which was routine for SMART[25].”
Revised: “In addition, while the total prescribed BED10 was similar for SMART and HART it is possible that our LF outcomes may have been influenced by radiobiologic differences related to fraction size; enhanced indirect cell killing mechanisms such as endothelial cell apoptosis mediated by the sphingomyelin pathway and tumor vascular damage may be triggered by fraction sizes of at least 10 Gy, which was routine for SMART[23].”
Comments 5: The Kaplan-Meier analyses for OS and PFS are presented with minimal interpretation. Given the borderline p-value for OS (P=0.056), a more thorough statistical assessment discussing clinical significance and sample size implications would be beneficial.
Response 5: There are limited conclusions that can be drawn regarding observed differences in OS. We acknowledged in the Discussion that “it is possible that we did not detect an OS differences as various patient, tumor, and treatment factors also impact OS.” It would be not appropriate for us to make any definitive statements about the impact of SMART vs HART technique on OS based on our retrospective analysis, and as we stated in the Conclusions we believe that these data “should be considered hypothesis-generating and support future prospective evaluation.”
Reviewer 2 Report
Comments and Suggestions for Authors
The manuscript entitled " Multi-institutional comparison of ablative 5-fraction magnetic resonance-guided online adaptive versus 15/25-fraction computed tomography-guided moderately hypofractionated offline adapted radiation therapy for locally advanced pancreatic cancer" was reviewed.
I found the retrospective data to be very interesting and meaningful. Based on that, I have the following questions: Currently, introducing chemotherapy prior to surgery is leading to continuous improvements in surgical outcomes. I believe this trial is important. Would patients who do not choose chemotherapy plus surgery be eligible for this trial? What are the reasons for choosing these treatment methods over current surgical therapy? Are the target patients simply those who do not want surgery? How do these treatments compare to surgery in terms of advantages?
Why does SMART produce poor results even though it has fewer local failures than HART?
The two-year overall survival rate in this study is nearly the same as that of carbon ion radiotherapy (C-ion RT). Does this indicate the limitations of radiation therapy? How can we improve these treatments to achieve even better outcomes in the future? Could you share any hopeful projections or possibilities? 
Author Response
The manuscript entitled " Multi-institutional comparison of ablative 5-fraction magnetic resonance-guided online adaptive versus 15/25-fraction computed tomography-guided moderately hypofractionated offline adapted radiation therapy for locally advanced pancreatic cancer" was reviewed.I found the retrospective data to be very interesting and meaningful. Based on that, I have the following questions:
Comments 1: Currently, introducing chemotherapy prior to surgery is leading to continuous improvements in surgical outcomes. I believe this trial is important. Would patients who do not choose chemotherapy plus surgery be eligible for this trial?
Response 1: We performed a retrospective analysis comparing two ablative radiation delivery techniques (SMART vs. HART) for patients with locally advanced pancreatic cancer who were not eligible surgery.
Comments 2: What are the reasons for choosing these treatment methods over current surgical therapy? Are the target patients simply those who do not want surgery? How do these treatments compare to surgery in terms of advantages?
Response 2: All patients were evaluated for surgery prior to receiving radiation therapy, although none was eligible for surgery because tumor was locally advanced. Describing the outcomes of ablative SMART or HART versus surgery is outside the scope of this analysis.
Comments 3: Why does SMART produce poor results even though it has fewer local failures than HART?
Response 3: Assuming that the reviewer is referring to OS differences between SMART and HART, we previously stated in the Discussion that “it is possible that we did not detect an OS differences as various patient, tumor, and treatment factors also impact OS.” OS is impacted by many factors other than LF.
Comments 4: The two-year overall survival rate in this study is nearly the same as that of carbon ion radiotherapy (C-ion RT). Does this indicate the limitations of radiation therapy? How can we improve these treatments to achieve even better outcomes in the future? Could you share any hopeful projections or possibilities? 
Response 4: Carbon-ion RT outcomes have been favorable, but comparing SMART or HART outcomes is outside the scope of this study. There certainly is a need to improve outcomes for PDAC patients although we believe that discussing other novel treatments in our manuscript would distract from the primary focus of analysis which is comparing two novel RT techniques.